# REDUCING OVER-CONFIDENT ERRORS OUTSIDE THE KNOWN DISTRIBUTION

## ABSTRACT

Intuitively, unfamiliarity should lead to lack of confidence. In reality, current algorithms often make highly confident yet wrong predictions when faced with unexpected test samples from an unknown distribution different from training. Unlike domain adaptation methods, we cannot gather an "unexpected dataset" prior to test, and unlike novelty detection methods, a best-effort original task prediction is still expected. We compare a number of methods from related fields such as calibration and epistemic uncertainty modeling, as well as two proposed methods that reduce overconfident errors of samples from an unknown novel distribution without drastically increasing evaluation time: (1) $\mathcal{G}$-distillation, training an ensemble of classifiers and then distill into a single model using both labeled and unlabeled examples, or (2) NCR, reducing prediction confidence based on its novelty detection score. Experimentally, we investigate the overconfidence problem and evaluate our solution by creating "familiar" and "novel" test splits, where "familiar" are identically distributed with training and "novel" are not. We discover that calibrating using temperature scaling on familiar data is the best single-model method for improving novel confidence, followed by our proposed methods. In addition, some methods' NLL performance are roughly equivalent to a regularly trained model with certain degree of smoothing. Calibrating can also reduce confident errors, for example, in gender recognition by 95% on demographic groups different from the training data.

## 1 INTRODUCTION

In machine learning and computer vision, the i.i.d. assumption, that training and test sets are sampled from the same distribution (henceforth "familiar" distribution), is so prevalent as to be left unwritten. In experiments, it is easy to satisfy the i.i.d. condition by randomly sampling training and test data from a single pool, such as photos of employees' faces. But in real-life applications, test samples are often sampled differently (e.g., faces of internet users) and may not be well-represented, if at all, by the training samples.

Prior work (Lakshminarayanan et al., 2017) has shown networks to be unreliable when tested on semantically unrelated input (e.g. feeding CIFAR into MNIST-trained networks), but users would not expect useful predictions on these data. However, we find this issue extends to *semantically related input* as well, such as gender classifiers applied to faces of older or younger people than those seen during training, which is a more common occurrence in practice and more problematic from a user's perspective. We demonstrate that, counter to the intuition that unfamiliarity should lead to lack of confidence, current algorithms (deep networks) are more likely to make highly confident wrong predictions when faced with such "novel" samples, both for real-world image datasets (Figure 1; see caption) and for toy datasets (Figure 2; see subcaption 2(a) and Appendix A). The reason is simple: the classification function, such as a deep network, is undefined or loosely regulated for areas of the feature space that are unobserved in training, so the learner may extrapolate wildly without penalty.

Confident errors on novel samples can be catastrophic. Whether one would ride in a self-driving car with a 99.9% accurate vision system, probably depends on how well-behaved the car is on the 0.1% mistakes. When a trained model labeled a person as a gorilla (Zhang, 2015), the public trust in that system was reduced. When a driving vision system confidently mistook a tractor trailer (Yadron & Tynan, 2016), a person died. Scholars that study the impact of AI on society consider differently distributed samples to be a major risk (Varshney & Alemzadeh, 2017): "This is one form of epistemic

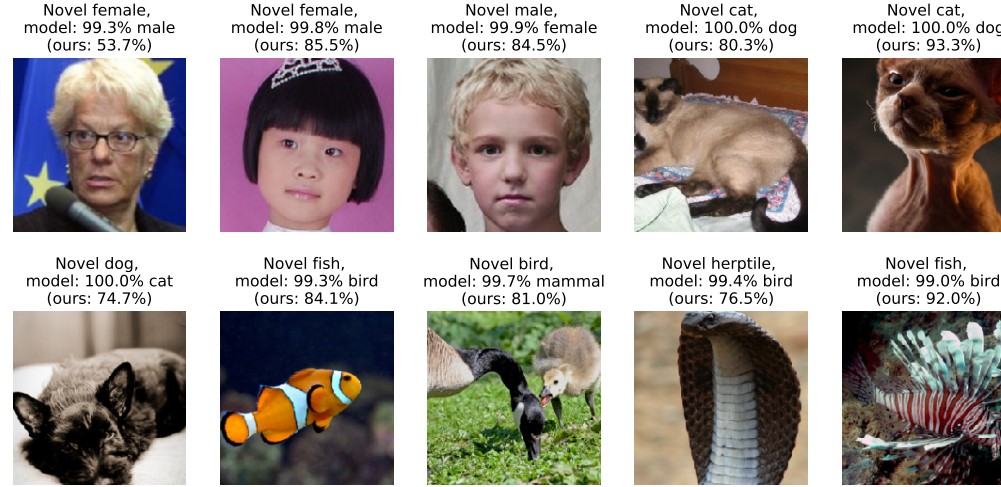

Figure 1: Deep networks can often make highly confident mistakes when samples are drawn from outside the distribution observed during training. Shown are example images that have ages, breeds, or species that are not observed during training and are misclassified by a deep network model with high confidence. Using our methods (shown here is $\mathcal{G}$-distillation, to distill an ensemble of networks on both training and unsupervised examples) makes fewer confident errors for novel examples, increasing the reliability of prediction confidence overall.

uncertainty that is quite relevant to safety because training on a dataset from a different distribution can cause much harm." Our trust in a system requires its ability to avoid confident errors.

Unfortunately, these novel samples may differ from training in both expected and unexpected ways. This means gathering a set of "unexpected" training samples, though required by covariate shift and domain adaptation methods, becomes unviable (Varshney & Alemzadeh, 2017). One may use novelty detection methods to identify novel samples at test time (to report an error or seek human guidance), but this may be insufficient. These "outliers" (e.g. underrepresented faces that users upload) are perfectly normal samples in the eyes of a user and they would reasonably expect a prediction. Also, human guidance may not be fast enough or affordable.

In this paper, our aim is to reduce confident errors for predictions on samples from a different (often non-overlapping) but unknown distribution (henceforth "novel" distribution). In contrast with most recent work, we focus on the *confidence of the prediction* rather than only the most likely prediction (accuracy) or the confidence ordering (average precision or ROC). In addition to reviewing the effectiveness of established methods, we propose and evaluate two ideas that are straightforward extensions to ensemble and rejection methods. One is that multiple learners, with different initializations or subsamples of training data, may make different predictions on novel data (see Lakshminarayanan et al. (2017)). Hence, ensembles of classifiers tend to be better behaved. But ensembles are slow to evaluate. If we distill (Hinton et al., 2015) an ensemble into a single model using the training data, the distilled classifier will have the original problem of being undefined for novel areas of the feature space. Fortunately, it is often possible to acquire many unlabeled examples, such as faces from a celebrity dataset. By distilling the ensemble on both the training set and on unsupervised examples, we can produce a single model that outperforms, in terms of prediction confidence, single and standard distilled models on both identically and differently distributed samples.

Another idea is that if the training set does not provide enough information for the unseen data, therefore it may be desired to simply avoid confident predictions. We can lower their confidence according to the output of an off-the-shelf novelty detector. This means reducing confident errors by reducing confidence, regardless of correctness. It may improve novel sample prediction quality, but in turn degrade performance on familiar samples. Although this idea sounds like a natural extension to novelty detection, we are unaware of any implementation or analysis in the literature.

Experimentally, we investigate the confidence problem and perform an extensive study by creating "familiar" and "novel" test splits, where "familiar" are identically distributed with training and

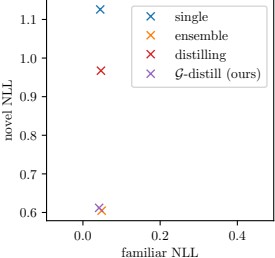

(a) Demonstration of deep networks' generalization behavior with a 2-dimensional toy dataset "square". **Column 1**: we design a binary ground truth for classification on a 2-dimensional feature space. **Column 2**: for training, we only provide samples on the left and lower portions (the "familiar" distribution), and reserve the upper-right only for testing (the "novel" distribution). **Column 3-7**: we show negative log likelihood (NLL) predicted for each point in the feature space. A small NLL indicates correct prediction, while a very large NLL indicates a confident error. **Column 3, 4**: multiple runs of the network have similar performances on familiar regions but vary substantially in novel regions where the training data imposes little or no regulation, due to optimization randomness. **Column 5**: an ensemble of 10 such networks can smooth the predictions and reduce confident errors at the sacrifice of test efficiency. **Column 6**: distilling the ensemble using the training samples results in the same irregularities as single models. **Column 7**: one of our proposals is to distill the ensemble into a single model using both the labeled training data and unsupervised data from a "general" distribution.

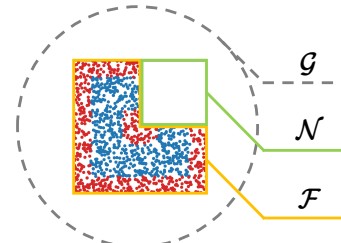

(b) Comparing familiar & novel negative log likelihood performance on toy dataset "square"

(c) Feature space partitioning: "familiar" $\mathcal{F}$, "novel" $\mathcal{N}$, and "general" $\mathcal{G}$ distributions. (see Appendix A and section 3)

Figure 2: This toy experiment illustrates how classifiers (in this case deep networks) can be confidently wrong when attempting to generalize outside the extent of the training data. See Appendix A for details and extra toy datasets. Figure best read in color.

"novel" are not. For example, in cat vs. dog classification, the novel examples are from breeds not seen during training, or in gender classification, the novel examples are people that are older or younger than those seen during training. Our evaluation focuses on negative log likelihood (NLL) and the fraction of highly confident predictions that are misclassified ("E99"). They measure both prediction accuracy and how often confident errors occur. To summarize, our contributions are:

- Draw attention to a counter-intuitive yet important problem of highly confident wrong predictions when samples are drawn from a unknown distribution that is different than training.

- Evaluate and compare the effectiveness of methods in related fields, including two proposed methods to reduce such overconfident errors.

- Propose an experimental methodology to study the problem by explicitly creating familiar and novel test splits and measuring performance with NLL and E99.

## 2 RELATED WORK

To the best of our knowledge, there is no prior work with exactly the same goal and focus as this paper, but we are very similar to several lines of work. To avoid confusion, we compile Table 1 to show our differences from these prior work for your reading convenience.

**Epistemic uncertainty.** The most related works model and reduce epistemic uncertainty (model uncertainty from incomplete knowledge) by estimating and evaluating a set or distribution of models that agree with the training data. The intuition is that on familiar test data, performance is boosted

Table 1: Difference between the goal of our method and several similar lines of research.

|  | assumes (or robust to) different train & test distr.? | novel distr. unknown during training? | novel samples in testing: |
|---|---|---|---|
| Ours (reduce confident errors) | Yes | Yes | Predict |
| Novelty & outlier detection | Yes | Varies | Reject |
| Rejection / failure detection | No | - | - |
| Semi-supervised learning | No | - | - |
| Epistemic uncertainty estimation | Yes | Yes | No evaluation |
| Active learning | Varies | No | Predict |
| Domain generalization | Yes | Limited[1] | Predict |
| Domain Adaptation | Yes | No | Predict |
| Calibration | No | - | - |

by aggregating outputs; but for novel data, the different models can disagree, indicating a lack of training evidence. In that case, aggregating the outputs results in an appropriately lower confidence.

Bayesian approaches (Bishop, 1995; Blundell et al., 2015; Hernández-Lobato et al., 2016) estimate a distribution of network parameters and produce a Bayesian estimate for likelihood. These methods are usually very computationally intensive (Lakshminarayanan et al., 2017), limiting their practical application. Gal & Ghahramani (2016) propose MC-dropout as a discrete approximation of Bayesian networks, adopting the dropout technique for a Monte-Carlo sample of likelihood estimates. They further propose to jointly estimate aleatoric and epistemic uncertainties (Kendall & Gal, 2017) to increase the performance and quality of uncertainty. Lakshminarayanan et al. (2017) propose to instead use an ensemble to emulate a Bayesian network and achieve better performance.

These works reduce uncertainty and improve generalization under i.i.d. assumptions. Our work differs in its focus of improving confidence estimates for novel samples (differently distributed from training), and our method is much more efficient at test time than MC-dropout or ensembles.

Hafner et al. (2018) reduces the confidence towards a prior on unseen data, which is similar to our proposed NCR method. We note that they perform regression on a low-dimensional feature space generalizing into future data, which is a very different problem from our image classification generalizing into unexpected samples. They use perturbed original dataset features and the prior as the labels for reducing confidence. This can be hard to generalize to an 224x224x3 image space.

**Domain adaptation** (Quionero-Candela et al., 2009) aims at training on a source domain and improving performance on a slightly different target domain, either through unsupervised data or a small amount of supervised data in the target domain. In our settings, it is unviable to sample a representative "unexpected dataset" prior to test time, which we consider unknowable in advance. Consequently, the unsupervised samples we use are *not* from the familiar or novel distribution.

**Domain generalization** (Muandet et al., 2013; Li et al., 2017; Shankar et al., 2018) is more closely related to our work, aiming to build models that generalize well on a previously unspecified domain, whose distribution can be different from all training domains. Unlike in domain adaptation, the target domain in this case is unknown prior to testing. These models generally build a domain-invariant feature space (Muandet et al., 2013) or a domain-invariant model (Shankar et al., 2018), or factor models into domain-invariant and domain-specific parts (Li et al., 2017).

Also related are attribute-based approaches, such as Farhadi et al. (2009), who build an attribute classifier that generalizes into novel object categories, similar to our experiment set-up. They select features that are discriminative for an attribute within individual classes to build invariance across object classes. These models all require multiple training domains to learn invariant representations, with the intent to improve robustness to variations in the target domain. In contrast, our method is concerned only with novel data and does not require multiple training domains.

**Some methods refuse to predict** by estimating whether the system is likely to fail and outputting a signal requesting external intervention when necessary. Meta-recognition (Scheirer et al., 2011) estimates the performance based on the model output. **Rejection options** (Fumera & Roli, 2002; Geifman & El-Yaniv, 2017) or failure detection (Zhang et al., 2014; Wang & Vasconcelos, 2018)

---

[1]Domain generalization requires several source domains, providing hints of the novel distribution (see text).

estimate the risk of misclassification to determine whether to refuse prediction, usually by looking at how close samples are to decision boundaries. Unlike our work, these works are not concerned with the quality of confidence estimates and do not analyze their rejection ability for samples from novel distributions.

**Outlier detection** (Devarakota et al., 2007; Lee & Grauman, 2012; Liang et al., 2018)**, novelty detection, and one-class classification** (Tax, 2001; Khan & Madden, 2009) determine whether a test sample comes from the same distribution as the training data. The downside is that these methods would provide no confidence estimate for rejected samples, even though the models can still provide informative estimates (e.g. in Section 5 gender classification, baseline accuracy on novel data is $85\%$ despite bad NLL performance). One of our methods naturally extends these methods by using their output to improve confidence estimates.

**Generalization.** Various techniques have been proposed in deep learning literature to minimize the generalization gap, and popular ones include data augmentation, dropout (Srivastava et al., 2014), batch normalization (Ioffe & Szegedy, 2015), and weight decay. Hoffer et al. (2017) propose better hyperparameter selection strategies for better generalization. Bagging (Breiman, 1996) and other model averaging techniques are also used prior to deep learning. These methods focus on reducing generalization gap between training and validation. They do not address issues with unexpected novel samples and can be used in conjunction with our method.

Theoretical analyses for off-training-set error (Schaffer, 1994; Roos et al., 2006) and empirical analysis of generalization for test samples (Novak et al., 2018) are also available, but these methods measure, rather than reduce, generalization errors.

**Calibration** methods (e.g. Guo et al. (2017)) aim to improve confidence estimates, but since the confidence estimates are learned from familiar samples (i.i.d. with training), risk is not reduced for novel samples. However, we experimentally find that (Guo et al., 2017) performs surprisingly well on unseen novel data, which the method is not optimized for.

**Distilling** (Hinton et al., 2015) can be used on unsupervised samples. Radosavovic et al. (2017) obtain soft labels on transformed unlabeled data and use them to distill for unsupervised learning. Li & Hoiem (2017) extend models to new tasks without retaining old task data, using the new-task examples as unsupervised examples for the old tasks with a distillation loss. Distillation has also been used to reduce sensitivity to adversarial examples that are similar to training examples (Papernot et al., 2016). Our work differs from all of these in the focus on producing accurate confidences for samples from novel distributions that may differ significantly from those observed in training. **One-shot learning** (e.g. Vinyals et al. (2016)) and **zero-shot learning** (e.g. Zendel et al. (2017)) aim to build a classifier through one sample or only metadata of the class. They focus on building a new class, while we focus on generalizing existing classes to novel samples within.

## 3 PROPOSED METHODS

The goal of this paper is to improve confidence estimate of deep models on unseen data. We focus on a classification setting, although our framework could be naturally extended to tasks similar to classification, such as VQA and semantic segmentation. We assume the probability of label given data $P(y|\boldsymbol{x})$ is the same where familiar and novel distributions overlap, but unlike covariate shift, we assume no knowledge of the novel distribution other than what is already in the familiar distribution.

**Notations.** Denote by $(\boldsymbol{x}_\mathcal{F}, y_\mathcal{F}) \sim \mathcal{F}$ the familiar data distribution, and $F_{tr}$ the training set, $F_{ts}$ the test set drawn from $\mathcal{F}$. Further denote by $(\boldsymbol{x}_\mathcal{N}, y_\mathcal{N}) \sim \mathcal{N}$ a novel data distribution, which satisfies $P_\mathcal{F}(y|\boldsymbol{x}) = P_\mathcal{N}(y|\boldsymbol{x})$ where the input distributions overlap. Denote by $N_{ts}$ the test set drawn from $\mathcal{N}$. The inputs $\boldsymbol{x}_\mathcal{F}$ and $\boldsymbol{x}_\mathcal{N}$ may occupy different portions of the entire feature space, with little to no overlap. Later in this section, we introduce and describe an unsupervised distribution $\boldsymbol{x}_\mathcal{G} \sim \mathcal{G}$ with training set $G_{tr}$.

In our problem setting, the goal is to improve performance and quality of the estimation for $P_\mathcal{N}(y|\boldsymbol{x})$. Only $F_{tr}$ (and unsupervised $G_{tr}$) are provided in training, while $F_{ts}$ and $N_{ts}$ are used in test time. No training sample from $\mathcal{N}$ is ever used, and $\mathcal{N}$ should not have been seen by the model, even during pre-training.

**Distillation of an ensemble.** We base our first method on the original distillation from an ensemble (Hinton et al., 2015), which we briefly summarize. First, train an ensemble $f_{Ens}(\cdot)$ on $F_{tr}$, which

Table 2: Illustration of the data usage of $\mathcal{G}$-distillation on familiar data $\mathcal{F}$, general data $\mathcal{G}$, and some novel data $\mathcal{N}$ that we assume no knowledge of.

| | Familiar $\mathcal{F}$ | General $\mathcal{G}$ | Novel $\mathcal{N}$ |
|---|---|---|---|
| training ensemble | $\mathcal{L}_{cls}$ w/ label $y_{\mathcal{F}}$ | – | – |
| distillation | $\mathcal{L}_{cls}$ w/ label $y_{\mathcal{F}}$ 
 $\mathcal{L}_{dis}$ w/ soft label $\widetilde{\boldsymbol{y}}_{\mathcal{F}}$ | – 
 $\mathcal{L}_{dis}$ w/ soft label $\widetilde{\boldsymbol{y}}_{\mathcal{G}}$ | – |
| evaluation | test on $F_{ts}$ | – | test on $N_{ts}$ |

consists of several networks such as ResNet18 (He et al., 2016). Then, for each $(\boldsymbol{x}_{\mathcal{F}}, y_{\mathcal{F}}) \in F_{tr}$, obtain a soft label $\widetilde{\boldsymbol{y}}_{\mathcal{F}}$, where $\widetilde{y}_{\mathcal{F}}^{(c)} = P(y = c | \boldsymbol{x}_{\mathcal{F}}; f_{Ens})$ is the probability estimate for class $c$ given by the ensemble. Finally, train a new single-model network $f_\theta$ by taking its probability prediction $\hat{\boldsymbol{y}}_{\mathcal{F}}$ and applying the distillation loss $\mathcal{L}_{dis}(\widetilde{\boldsymbol{y}}_{\mathcal{F}}, \hat{\boldsymbol{y}}_{\mathcal{F}})$. This is simply a cross-entropy loss between the distribution estimates, with a temperature $T$ applied to the logits (see Hinton et al. (2015)) to put more focus on relative probability mass.

To further improve distillation performance, a classification loss $\mathcal{L}_{cls}$ (cross-entropy is used) over $\mathcal{F}$ is added as an auxiliary. The final loss becomes:

$$\mathcal{L} = \frac{1}{|F_{tr}|} \sum_{(\boldsymbol{x}_{\mathcal{F}}, y_{\mathcal{F}}) \in F_{tr}} \left( \lambda_{cls} \mathcal{L}_{cls}(y_{\mathcal{F}}, \hat{\boldsymbol{y}}_{\mathcal{F}}) + \lambda_{dis} \mathcal{L}_{dis}(\widetilde{\boldsymbol{y}}_{\mathcal{F}}, \hat{\boldsymbol{y}}_{\mathcal{F}}) \right). \quad (1)$$

As can be seen, distillation is still a method focused on the familiar distribution $\mathcal{F}$, and we have shown that a distilled network is not necessarily well-behaved on $\mathcal{N}$.

**Method 1: $\mathcal{G}$-distillation of an ensemble with extra unsupervised data.** To improve distillation on novel data, a natural idea would be having the distilled model mimic the ensemble on *some kind of* novel data. Denote by $\boldsymbol{x}_{\mathcal{G}} \sim \mathcal{G}$ an unlabeled general data distribution which *ideally* encompasses familiar $\mathcal{F}$ and any specific novel $\mathcal{N}$. Here "encompass" means that data from $\mathcal{F}$ and $\mathcal{N}$ can appear in $\mathcal{G}$: $\forall \boldsymbol{x}$, $P_{\mathcal{G}}(\boldsymbol{x}) \gg 0$ if $P_{\mathcal{F}}(\boldsymbol{x}) \gg 0$ or $P_{\mathcal{N}}(\boldsymbol{x}) \gg 0$. We draw a training set $G_{tr}$ from such general distribution $\mathcal{G}$.

A distribution that has all related images is nearly impossible to sample in practice. Hence, we cannot rely on $G_{tr} \sim \mathcal{G}$ to encompass all possible novel $\mathcal{N}$. We need to pick a dataset that is sufficiently diverse, sufficiently complex, and sufficiently relevant to the task, so our method can use it to generalize. Often, such a set can be obtained through online searches or mining examples from datasets with different annotations. For example, for making a gender classifier robust to additional age groups, we could sample $G_{tr}$ from the CelebA (Liu et al., 2015) dataset (note that CelebA does not have age information and we do not need it; for the sake of the experiment we also discard gender information from CelebA). For making a cat-dog classifier robust to novel breeds, we could sample $G_{tr}$ from ImageNet (Russakovsky et al., 2015). Note these $G_{tr}$'s do not encompass $\mathcal{N}$, but only provide a guideline for generalization.

Similar to distillation, we train an ensemble $f_{Ens}$ and obtain the soft labels $\widetilde{\boldsymbol{y}}_{\mathcal{F}}$. In addition, we also obtain soft labels $\widetilde{\boldsymbol{y}}_{\mathcal{G}}$ by evaluating $f_{Ens}$ on $\boldsymbol{x}_{\mathcal{G}} \in G_{tr}$. Then we add the samples $G_{tr}$ into the training set, and train using the combined data:

$$\mathcal{L} = -\frac{1}{|F_{tr} \cup G_{tr}|} \left( \sum_{G_{tr}} \lambda_{dis} \mathcal{L}_{dis}(\widetilde{\boldsymbol{y}}_{\mathcal{G}}, \hat{\boldsymbol{y}}_{\mathcal{G}}) + \sum_{F_{tr}} \left( \lambda_{dis} \mathcal{L}_{dis}(\widetilde{\boldsymbol{y}}_{\mathcal{F}}, \hat{\boldsymbol{y}}_{\mathcal{F}}) + \lambda_{cls} \mathcal{L}_{cls}(y_{\mathcal{F}}, \hat{\boldsymbol{y}}_{\mathcal{F}}) \right) \right) \quad (2)$$

Note that $\mathcal{G}$ is unsupervised, so we cannot apply $\mathcal{L}_{cls}$ on its samples. For test time, we simply evaluate the probability estimation $\widetilde{\boldsymbol{y}}_{\mathcal{F}}$ against the ground truth in $F_{ts}$, and $\widetilde{\boldsymbol{y}}_{\mathcal{N}}$ against those in $N_{ts}$, respectively. Table 2 demonstrates our training and evaluation procedure.

**Method 2: Novelty Confidence Reduction (NCR) with a novelty detector.** This method is more straightforward. We make use of a recent off-the-shelf, model-free novelty detector ODIN (Liang et al., 2018). We run both the single model and the ODIN procedure on the input $\boldsymbol{x}$ to get the probability estimate $\hat{\boldsymbol{y}}$ and the detection score $s_0(\boldsymbol{x})$ (where a *smaller* $s_0$ means a more novel input).

Table 3: Dataset usage. Each dataset is divided into $\mathcal{F}$ and $\mathcal{N}$ portions, and gereral $\mathcal{G}$ is selected by trying to choose a dataset diverse enough to encompass samples from $\mathcal{F}$ and potential $\mathcal{N}$ in the wild. (*) Please see text for details.

| Task | familiar $\mathcal{F}$ | novel $\mathcal{N}$ | subsample | general $\mathcal{G}$ |
|---|---|---|---|---|
| Gender recognition | LFW+ (Han & Jain, 2014) age 18-59 (Han et al., 2017) | LFW+, age 0-17 & 60+ | – | CelebA (Liu et al., 2015) |
| ImageNet superclasses | ILSVRC12, some species* | ILSVRC12, other species* | Per superclass: train: 1k, test: 400 | COCO (Lin et al., 2014) |
| Cat-dog binary recognition | Pets, some breeds* (Parkhi et al., 2012) | Pets, other breeds* | – | ILSVRC12 (Russakovsky et al., 2015) |
| VOC-COCO recognition | VOC, 20 classes (Everingham et al., 2015) | COCO, but ignore non-VOC classes | – | Places365-standard (Zhou et al., 2017) |

However, $s_0$ may not be calibrated and may lie in an arbitrary interval specific to the detector, e.g. $[0.5, 0.51]$. We normalize it to $[0,1]$ with a piecewise-linear function $s(\boldsymbol{x}) = \max\left(0, \min\left(\frac{s_{95\%}-s_0(\boldsymbol{x})}{s_{95\%}-s_{5\%}}, 1\right)\right)$ where $s_{\cdot\%}$ are the $\cdot\%$ percentiles of the detection scores for all training samples. $s(\boldsymbol{x})$ closer to 1 means the confidence should be reduced more.

At test time, we linearly interpolate between $\hat{\boldsymbol{y}}$ and the prior $\boldsymbol{y}_0$:

$$\hat{\boldsymbol{y}}_{NCR} = (1 - \lambda_s s(\boldsymbol{x}))\hat{\boldsymbol{y}} + \lambda_s s(\boldsymbol{x})\boldsymbol{y}_0 \tag{3}$$

where $y_0^{(c)} = P_{\mathcal{F}}(y = c)$ is the prior of class $c$ on the familiar $F_{tr}$, and $\lambda_s = 0.15$ a hyperparameter.

**Efficiency comparison.** An ensemble (Lakshminarayanan et al., 2017) requires $M$ forward passes where $M$ is the number of ensemble members. MC-dropout (Gal & Ghahramani, 2016) requires $\sim 50\times$ forward passes for the Monte Carlo samples. $\mathcal{G}$-distillation needs only one forward pass, and NCR only needs to evaluate an extra novelty detector (in particular, ODIN needs a forward-backward pass). Therefore at test time they are much faster. However, $\mathcal{G}$-distillation pays a price at training time by training both an ensemble and a distilled model.

We refer readers to Appendix B for implementation details.

## 4 EXPERIMENTAL SETUP

**Datasets.** To demonstrate effectiveness in appropriately handling unexpected novel data, and reduce confident failures thereof, we perform extensive analysis on four classification datasets mimicking different scenarios. We set up the $\mathcal{F}$, $\mathcal{N}$, and $\mathcal{G}$ distributions, and make sure that $\mathcal{F}$ and $\mathcal{N}$ are completely non-overlapping, unless otherwise noted. Table 3 illustrates our datasets:

- Gender recognition, mimicking a dataset with obvious selective bias.
- ImageNet animal superclass (mammals, birds, herptiles, and fishes) recognition, mimicking an animal classifier being tested on unseen species within these superclasses. (*) We determine the original classes belonging to each superclass, sort the classes by their indices, and use the first half of the class list as familiar and the second half novel. This makes sure $\mathcal{F}$ and $\mathcal{N}$ do not overlap by definition.
- Cat-dog recognition. Similar in spirit as the above. (*) The first 20 dog breeds and 9 cat breeds are deemed familiar, and the rest are novel.
- VOC-COCO recognition (determining presence of object categories). Mimics a model trained on a lab dataset being applied on more complex real world images with unexpected input. `tvmonitor` is mapped to `tv` and not `laptop`.

Note that VOC-COCO is quite different from the others where $\mathcal{F}$ and $\mathcal{N}$ do not overlap, because VOC and COCO images can be very similar.

**Pre-training.** When designing our experiments, we should make sure $\mathcal{N}$ is still novel even considering information gained from the pre-training dataset. Although using ImageNet (Russakovsky et al., 2015) pre-trained models may improve performance, we note that the dataset almost always contains classes that appear in our "unseen" novel $\mathcal{N}$ choices; therefore pre-training on ImageNet would render the novel $\mathcal{N}$ not actually novel. Instead, we opt to use Places365-standard (Zhou

et al., 2017) as the pre-training dataset. We argue that our method would generalize to ImageNet pre-trained models when $\mathcal{N}$ classes are disjoint from ImageNet.

**Validation splits.** Since we need to report negative log likelihood on both $F_{ts}$ and $N_{ts}$, datasets with undisclosed test set cannot be directly used. For ImageNet animals, VOC, and COCO, we report performance on the validation set, while splitting the training set for hyperparameter tuning. For LFW+ and Oxford-IIIT Pets, we split the training set for validation and report on the test set. For LFW+, we use the first three folds as training, and the last two folds as test since the dataset is smaller after the familiar-novel split. For the general $G_{tr} \sim \mathcal{G}$ dataset we proportionally sample from the corresponding training set, or a training split of it during hyperparameter tuning.

**Compared methods.** We compare to normal single models ("single"), and standard distillation using $F_{tr}$ images ("distilling") as baselines. We compare to using an ensemble ("ensemble"), which is less efficient in test time. In addition, we compare to Kendall & Gal (2017) where aleatoric and epistemic uncertainties are modeled to increase performance ("uncertainty"). Since ResNet18 does not use dropout, for this method we insert a dropout of $p = 0.2$ after the penultimate layer. For a fairer comparison, we also include an experiment using DenseNet161 as the base network, which is designed to use dropout. We also compare to (Guo et al., 2017) ("$T$-scaling") where only familiar samples are considered to calibrate the confidences, as well as an ensemble with members with the same calibration ("calib. ens.").

In our experiments, we discover that some methods outperform on novel samples and underperform on familiar, and one may suspect that the trade off is simply an effect of smoothing the probability regardless of the sample being novel or familiar, or the prediction being correct or wrong. To analyze this, we also report the performance of further raising or decreasing the temperature of the prediction logits of each method by $\tau$:

$$\hat{y}^{(c)\prime} = \frac{(y^{(c)})^{1/\tau}}{\sum_i (y^{(i)})^{1/\tau}} \tag{4}$$

where $\hat{y}^{(c)}$ is the original estimate for class $c$. We use $\tau \in [0.5, 5]$, and plot the trace for how the familiar and novel NLL change. Note that this is equivalent to using (Guo et al., 2017) with a number of different temperature choices, but the trace can be optimistic (and seen as an oracle), since in practice the choice of $\tau$ has to be decided before evaluating on test.

**Performance measure.** Note that we want to measure how well the model can generalize, and how unlikely the model is to produce confident errors. As discussed in Section 1, we mainly use the negative log likelihood (NLL), a long-standing performance metric (Nasrabadi, 2007; Friedman et al., 2001), as a measure of the quality of our prediction confidence. If a prediction is correct, we prefer it to be confident, leading to a lower NLL. And for incorrect predictions, we prefer it to be unconfident, which means the likelihood of the *correct class* should in turn be higher, which also leads to a lower NLL. The metric gives more penalty towards confident errors, suiting our needs. In summary, the model needs to learn from only familiar labeled data (plus unsupervised data) and produce the correct confidence (or lack of confidence) for novel data to improve the NLL.

During evaluation, we clip the softmax probability estimates given by all models to $[0.001, 0.999]$ in order to prevent a small number of extremely confident wrong predictions from dominating the NLL. We clip these estimates also considering overconfidence beyond this level is risky with little benefit in general. The processed confidences still sum up to 1 for binary classifiers, but may be off by a small amount for multiclass scenarios.

We also report a confident error rate ("E99") where we show the error rate for samples with probability estimate of any class larger than 0.99. Ideally this value should be within $[0, 0.01]$. Further, we also report accuracy and mean Average Precision (mAP) and test the hypothesis that we perform comparably on these metrics. Due to the relatively high variance of NLL on $N_{ts}$, we run our experiments 10 times to ensure statistical significance with a p-test, but we run the ensemble method only once (variance estimated using ensemble member performance variance). Our 10 runs of the distillation methods use the same ensemble run.

## 5 RESULTS

We first compare $\mathcal{G}$-distill to the baselines on the four scenarios. Tables in Figures 3 and 4 show our results, and the graphs show the trade-off between familiar and novel NLL by smoothing. Note

| | NLL | | E99 | | accuracy | |
|---|---|---|---|---|---|---|
| | familiar | novel | familiar | novel | familiar | novel |
| uncertainty | 0.080 | 0.567 | 0.005 | 0.068 | 0.974 | 0.853 |
| distilling | 0.077 | 0.364 | 0.001 | **0.002** | 0.973 | 0.844 |
| $T$-scaling | 0.080 | **0.328** | 0.001 | 0.003 | 0.972 | 0.853 |
| single | 0.083 | 0.542 | 0.005 | 0.060 | 0.972 | 0.853 |
| ensemble | **0.062** | 0.455 | 0.002 | 0.039 | 0.978 | 0.853 |
| calib. ens. | 0.074 | **0.313** | **0.000** | **0.001** | 0.979 | 0.852 |
| $\mathcal{G}$-distill (ours) | 0.073 | 0.337 | 0.001 | **0.001** | 0.975 | 0.847 |
| NCR (ours) | 0.089 | 0.334 | 0.001 | **0.001** | 0.972 | 0.854 |

| | NLL | | E99 | | accuracy | |
|---|---|---|---|---|---|---|
| | familiar | novel | familiar | novel | familiar | novel |
| uncertainty | 0.310 | 1.123 | 0.014 | 0.094 | 0.901 | 0.713 |
| distilling | 0.302 | 0.921 | 0.007 | 0.042 | 0.895 | 0.710 |
| $T$-scaling | 0.331 | 0.760 | **0.000** | **0.000** | 0.896 | 0.709 |
| single | 0.326 | 1.128 | 0.013 | 0.087 | 0.896 | 0.709 |
| ensemble | **0.256** | 0.930 | 0.005 | 0.045 | 0.906 | 0.724 |
| calib. ens. | 0.317 | **0.723** | **0.000** | **0.000** | 0.907 | 0.725 |
| $\mathcal{G}$-distill (ours) | 0.279 | 0.868 | 0.007 | 0.034 | 0.904 | 0.716 |
| NCR (ours) | 0.308 | 0.855 | **0.001** | 0.005 | 0.896 | 0.709 |

(a) Gender recognition with LFW+

(b) Animal recognition with ImageNet superclasses

| | NLL | | E99 | | accuracy | |
|---|---|---|---|---|---|---|
| | familiar | novel | familiar | novel | familiar | novel |
| uncertainty | 0.056 | 0.417 | 0.004 | 0.053 | 0.984 | 0.908 |
| distilling | 0.060 | 0.332 | 0.002 | 0.036 | 0.982 | 0.901 |
| $T$-scaling | 0.040 | 0.275 | 0.001 | 0.017 | 0.984 | 0.905 |
| single | 0.053 | 0.423 | 0.004 | 0.053 | 0.984 | 0.905 |
| ensemble | **0.033** | 0.286 | 0.001 | 0.024 | 0.988 | 0.912 |
| calib. ens. | **0.032** | **0.220** | **0.000** | 0.010 | 0.988 | 0.912 |
| $\mathcal{G}$-distill (ours) | 0.062 | 0.307 | 0.001 | 0.022 | 0.979 | 0.897 |
| NCR (ours) | 0.063 | 0.288 | **0.000** | **0.000** | 0.984 | 0.904 |

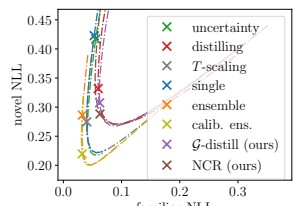

(c) Cat-dog recognition with Pets

Figure 3: Our performance on familiar $\mathcal{F}$ and novel data $\mathcal{N}$, for tasks with no overlap between the two. **Tables:** in terms of novel NLL, calibrated methods perform the best, while our methods outperform the rest of the single-model methods. Methods perform similarly on familiar NLL, while ours and calibration are more reliable with high-confidence estimates (E99), indicating a better generalization. We perform on the same level on accuracy or mAP as other methods except for $\mathcal{G}$-distill in (c). **Graphs:** Trade off familiar and novel NLL by smoothing the probability estimates with a higher or lower temperature. Crosses indicate performance without smoothing ($\tau = 1$). Bottom-left is better. Our methods do not outperform single models considering smoothing trade-off.

| | NLL | | E99 | | mAP | |
|---|---|---|---|---|---|---|
| | familiar | novel | familiar | novel | familiar | novel |
| uncertainty | 0.086 | 0.129 | 0.003 | 0.006 | 0.806 | 0.543 |
| distilling | 0.087 | **0.122** | 0.001 | 0.003 | 0.807 | 0.547 |
| $T$-scaling | 0.087 | 0.127 | 0.001 | 0.003 | 0.805 | 0.545 |
| single | 0.086 | 0.128 | 0.002 | 0.005 | 0.805 | 0.545 |
| ensemble | **0.082** | 0.123 | 0.002 | 0.005 | 0.817 | 0.559 |
| calib. ens. | 0.084 | 0.123 | **0.001** | **0.003** | 0.817 | 0.559 |
| $\mathcal{G}$-distill (ours) | 0.087 | 0.122 | 0.001 | 0.003 | 0.802 | 0.541 |
| NCR (ours) | 0.087 | 0.125 | 0.002 | 0.004 | 0.804 | 0.544 |

Figure 4: Our performance on VOC-COCO recognition, where familiar $\mathcal{F}$ and novel data $\mathcal{N}$ have a strong overlap. **Table:** for NLL, we outperform other single-model based methods on novel data but underperform on familiar. Calibration methods do not make much difference. **Graph:** considering trade-off between familiar and novel NLL, $\mathcal{G}$-distillation performs similarly to distillation, while NCR underperforms. See Figure 3 for details. Note that this figure is zoomed in more.

that among these experiments, those in Figure 3 have novel distribution $\mathcal{N}$ completely disjoint from familiar $\mathcal{F}$, while Figure 4 does not.

*Single models and the ensemble:* single models perform much worse on novel NLL than familiar. Their familiar E99 is properly small, but novel E99 is far beyond 0.01. This confirms that confident errors in novel data is indeed a problem in our real image datasets. Ensembles not only improve performance on familiar data, but also are much better behaved on novel data.

*Calibration:* Except in Figure 4, properly calibrated $T$-scaling with single models perform as well as, or better than, all methods using a single model at test time. An ensemble of calibrated members is nearly equivalent to a ensemble with further smoothed predictions, and it performs best in terms of novel NLL (at the expense of familiar NLL, and test efficiency). It is very surprising that even when calibrated using familiar data only, single models are able to outperform all methods except ensembles of calibrated members in Figure 3. These indicate that proper calibration is a key factor when reducing confident errors even on novel data. In Figure 4, calibration does not make much difference (single models are well-calibrated already).

*Our proposed methods:* $\mathcal{G}$-distill and sometimes NCR perform reasonably well in familiar-novel balance (except on Figure 3(c)). In Figure 4 (VOC-COCO), distilling and $\mathcal{G}$-distill outperforms on novel NLL, but note that the difference is much smaller compared to other experiments.

*With smoothing trade-off:* Looking at predictions with different levels of smoothing, we can tell that uncertainty, distillation, $\mathcal{G}$-distill, and NCR are all equivalent or inferior to a smoothed version of a single model. Considering smoothing with $\tau$ is equivalent to tuning $T$-scaling on test data, this means that (1) these methods may outperform single models in accuracy or mAP, and have better-calibrated NLL than single models, but are ultimately inferior to better calibrated single models in confidence quality; and (2) even if these methods are calibrated, their confidence may not outperform a calibrated single model trained regularly. We note that in practice, one cannot choose the smoothing $\tau$ using a test novel dataset, so the smoothing results are only for analysis purposes.

*Confident errors:* in Figure 3, our error rate among novel confident predictions is far below compared non-calibrated methods and much closer to familiar E99. For Figure 4, we also have a slightly lower novel E99. These indicate a better calibration with confident predictions, especially with a disjoint $\mathcal{N}$. However, the E99 for both NCR and the calibrated methods ($T$-scaling and ensemble with calibrated members) is usually far lower than 0.01, suggesting often under-confident predictions.

*Other metrics:* for accuracy or mAP, our methods (especially NCR) remain competitive compared to methods other than the ensemble, in Figures 3(a) and 3(b). However, they only perform similarly to distillation in Figure 3(c) and slightly underperform the others in Figure 4. Calibration with $T$-scaling and smoothing with temperature do not change accuracy or mAP.

*Novelty detector effectiveness:* one interesting phenomenon is that the novelty detector is not very effective in distinguishing familiar and novel samples (AUC for each dataset: LFW+ 0.562, ImageNet 0.623, Pets 0.643, VOC-COCO 0.665), but are quite effective in separating out wrongly classified samples (AUC: LFW+ 0.905, ImageNet 0.828, Pets 0.939, VOC-COCO 0.541). We hypothesize that the novelty detector can fail to detect some data in our novel split that are too similar to the familiar. However, this does not affect the prediction performance much since the classifier model are less prone to confident failures on these samples.

**Miscellaneous.** In Appendix C, we evaluate the impact of some factors on our experiments, namely the choice of ensemble type, and the base network size. It is also possible to train an ensemble of $\mathcal{G}$-distillation models to boost the performance, at the sacrifice of test time performance. We find that it improves the foreign NLL beyond the original ensemble, while still underperforming calibration.

We also tried to combine $\mathcal{G}$-distillation and NCR, by performing novelty detection on the $\mathcal{G}$-distilled network and further reduce novelty confidence. However, the results show that only when both methods show advantage against baselines, the combined method can outperform both components. Otherwise the combined method may underperform baselines.

# 6 CONCLUSIONS

In this paper, we draw attention to the importance of minimizing harm from confident errors in unexpected novel samples different from training. We propose an experiment methodology to explicitly study generalization issues with unseen novel data, and compare methods from several related fields. We propose two simple methods that use an ensemble and distillation to better regularize network behavior outside the training distribution, or reduce confidence on detected novel samples, and consequently reduce confident errors. For future work, it can be beneficial to investigate the ability to handle adversarial examples using this framework, and improve calibration with unexpected novel samples taken into account.

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

## A   Toy experiment details

To better visualize the problem of confident errors when using single neural networks, we demonstrate the generalization behavior of networks on 2-dimensional feature spaces. We construct toy datasets with input $x \in \mathbb{R}^2$ and define labels $y \in \{0, 1\}$. For each dataset we split the feature space into familiar region $\mathcal{F}$ and novel region $\mathcal{N}$, train models on $\mathcal{F}$ and evaluate on a densely-sampled meshgrid divided into $\mathcal{F}$ and $\mathcal{N}$ regions. See Figure 2, column 1 for an illustration of the ground truth and familiar set of our "square" dataset. Intuitively, the "square" dataset seems to be easy, and the network may perform well on the novel data, while for harder datasets the network might either perform badly or produce a low-confidence estimation.

Figures 2 and 5 show the results. In the "square" dataset, the networks do not generalize to the novel corner, but rather draw an irregular curve to hallucinate something smoother. As a consequence, the networks become confidently wrong on a significant portion of the dataset. In multiple runs of the optimization, this behavior is consistent, but the exact regions affected are different. The ensemble gives a much smoother result and (appropriately) less confident estimates on the novel region, and the area of confident errors is largely reduced.

Figure 2(b) further shows the negative log likelihood (NLL) performance of each method. While the performance for familiar samples is similar across methods, the NLL for novel samples improves dramatically when using an ensemble. This indicates that the ensemble has a more reliable confidence estimate on novel data.

We further test standard distillation (Hinton et al., 2015) and our method (see Section 3 for notations and definitions) against these baselines. We note that this experiment can only be seen as *an upper bound* of what our method could achieve, because when we sample our unsupervised set from the general distribution in this experiment, we use $\mathcal{F}$ and $\mathcal{N}$ combined for simplicity. In our later experiments on real images, the novel distribution is considered unknown.

Figure 2(b) shows that standard distillation performs not much better than single methods, consistent with our hypothesis that distillation using the familiar samples still results in irregularities on novel data. On the other hand, our method performs much closer to the ensemble. This positive result encourages us to further test our method on datasets with real images.

Figure 5 illustrates the irregularities of single models with more toy experiments, and shows the performance of all methods involved. Although the toy datasets can be very simple ("xor" and "circle") or very complex ("checkerboard"), in all cases, the same observations in Figure 2 applies. The single models are more likely to have erratic behaviors outside the familiar region, the ensemble behaves more regularly, and our method can emulate the ensemble in terms of the robustness in novel regions.

## B   Implementation details

**For toy experiments**. We take 1200 samples for both train and validation, while our test set is simply a densely-sampled meshgrid divided into familiar and novel test sets. We use a 3-hidden-layer network, both layers with 1024 hidden units and Glorot initialization similar to popular deep networks, to avoid bad local minima when layer widths are too small Choromanska et al. (2015). Batchnorm Ioffe & Szegedy (2015) and then dropout Srivastava et al. (2014) are applied after ReLU. We use the same hyperparameter tuning, initialization, and training procedures as described in Section 3 implementation details.

**For image experiments**. We use the following experimental settings, unless otherwise explained. We set $\lambda_{dis} = \frac{2}{3}$ and $\lambda_{cls} = \frac{1}{3}$. The original paper used 1 and 0.5, but we scale them to better compare with non-distillation methods. Temperature $T = 2$. We sample $G_{tr}$ to be roughly $\frac{1}{4}$ the size of $F_{tr}$. For the network, we mostly use ResNet18 (He et al., 2016), but we also perform an experiment on DenseNet161 (Huang et al., 2017) to check that our conclusions are not architecture-dependent. To enable training on small datasets, we use networks pre-trained on very large datasets as a starting point for both ensemble members and the distilled network. We use a simple ensemble of 10 members. We also compare to a bagging (Breiman, 1996) ensemble scheme which resamples the dataset. This is more popular prior to deep networks, but we empirically find that bagging undermines the log likelihood on both familiar data and novel data.

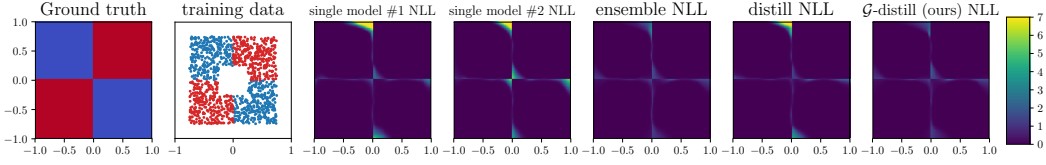

(a) Toy dataset "xor" and negative log likelihood across the feature space.

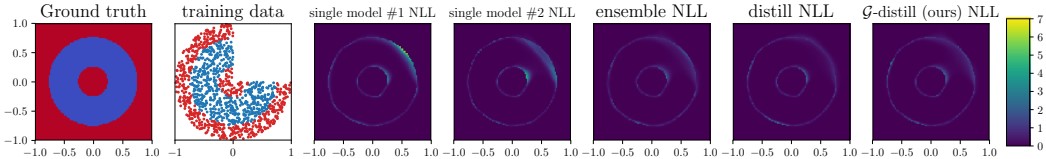

(b) Toy dataset "circle" and negative log likelihood across the feature space.

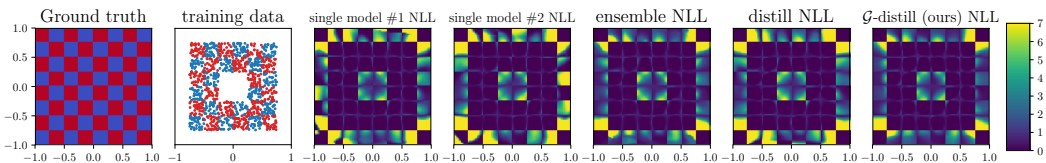

(c) Toy dataset "checkerboard" and negative log likelihood across the feature space.

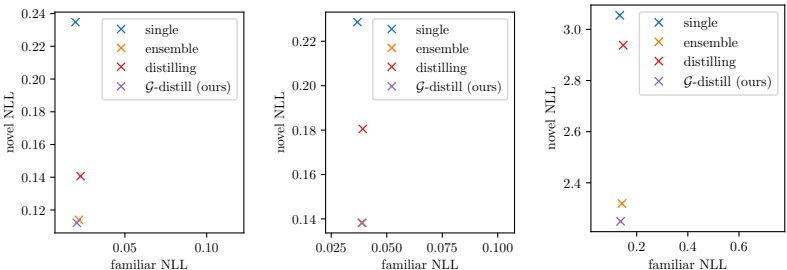

(d) Comparison of familiar and novel NLL for the three datasets. Left to right: "xor" "circle" and "checkerboard".

Figure 5: A continuation of the toy experiment in Figure 2 with more datasets. Classifiers (in this case deep networks) can be confidently wrong when attempting to generalize outside the extent of the training data. On these regions, an ensemble is better behaved and makes less confident mistakes. $\mathcal{G}$-distill is more able to mimic this behavior than standard distillation. All methods perform similarly in the familiar region. See Figure 2 for details. Figure best read in color.

We initialize the final layer of our pre-trained network using Glorot initialization (Glorot & Bengio, 2010). We optimize using stochastic gradient descent with a momentum of 0.9. For data augmentation, we use a random crop and mirroring similar to Inception (Szegedy et al., 2015). At test time we evaluate on the center crop of the image. We lower the learning rate to 10% when validation performance plateaus and run an additional 1/3 the number of past epochs.

We perform hyper-parameter tuning for e.g. the learning rate, the number of epochs, and $T$ in Guo et al. (2017) using a manual search on a validation split of the training data, but use these hyper-parameters on networks trained on both training and validation splits of the training data. Note that $\mathcal{N}$ is unknown while training, so we should use the $F_{ts}$ performance (accuracy, mAP) as the only tuning criteria. However, $\lambda_s$ for our second method NCR needs to be tuned according to some novelty dataset, which we assume unavailable. This makes a fair hyper-parameter tuning hard to design. We further split the Pets dataset familiar $F_{tr}$ into validation-familiar and validation-novel splits, again by assigning some breeds as familiar and others as novel. We tune the hyperparameter on this validation set with a grid search, and use the result $\lambda_s = 0.15$ on all experiments. We also manually tune the choice of percentiles (95% and 5%) this way.

| | NLL | | E99 | | accuracy | |
|---|---|---|---|---|---|---|
| | familiar | novel | familiar | novel | familiar | novel |
| simple distilling | 0.302 | 0.921 | 0.007 | 0.042 | 0.895 | 0.710 |
| simple ensemble | **0.256** | 0.930 | **0.005** | 0.045 | 0.906 | 0.724 |
| simple $\mathcal{G}$-distill | 0.279 | 0.868 | **0.007** | 0.034 | 0.904 | 0.716 |
| bagging distilling | 0.310 | 0.878 | 0.012 | **0.027** | 0.892 | 0.711 |
| bagging ensemble | 0.282 | 0.881 | 0.007 | 0.037 | 0.894 | 0.723 |
| bagging $\mathcal{G}$-distill | 0.291 | **0.844** | 0.011 | **0.024** | 0.898 | 0.715 |

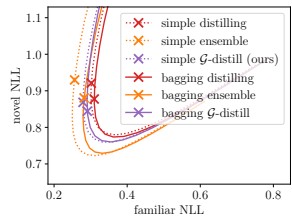

Figure 6: Performances on animal recognition with ImageNet using and distilling from a bagging ensemble instead. This reduces familiar performance.

| | NLL | | E99 | | accuracy | |
|---|---|---|---|---|---|---|
| | familiar | novel | familiar | novel | familiar | novel |
| uncertainty | 0.080 | 0.593 | 0.007 | 0.074 | 0.978 | 0.859 |
| distilling | 0.071 | 0.412 | 0.004 | 0.025 | 0.979 | 0.865 |
| $T$-scaling | 0.062 | 0.328 | 0.001 | 0.008 | 0.978 | 0.857 |
| single | 0.077 | 0.561 | 0.006 | 0.066 | 0.978 | 0.857 |
| ensemble | **0.051** | 0.410 | 0.002 | 0.035 | 0.982 | 0.866 |
| calib. ens. | 0.053 | **0.284** | **0.000** | **0.001** | 0.982 | 0.866 |
| $\mathcal{G}$-distill (ours) | 0.064 | 0.356 | 0.003 | 0.014 | 0.980 | 0.861 |
| NCR (ours) | 0.082 | 0.323 | 0.001 | **0.000** | 0.978 | 0.859 |

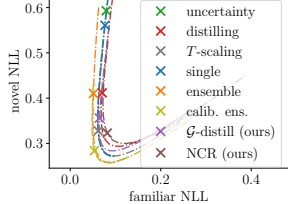

Figure 7: Performances on gender recognition with LFW+ using DenseNet161 as the backbone. Calibrated methods perform the best, and ours outperform other single-model-based methods.

## C EXTRA EXPERIMENTS

**Ensemble type.** We use simple ensembles since they are reported to perform better than a bagging ensemble (Lakshminarayanan et al., 2017) on the familiar dataset. We investigate whether bagging will benefit novel data in turn. We compare bagging to a simple ensemble and the distillation methods with them on the animal superclass task in Figure 6.

The simple ensemble indeed has better familiar performance, but without smoothing, bagging has better novel performance. Considering smoothing trade-off, there is a loss in both novel and familiar NLL when using a bagging ensemble.

**Network structure.** To evaluate the robustness of our experiments to the base network, we use a DenseNet161 trained on Places365-standard (provided by Zhou et al. (2017)) as the base model, and perform our experiment on the gender recognition task in Figure 7. Our observations from using ResNet18 hold. Our methods perform better than single models, but are outperformed by proper calibrated models. However, $\mathcal{G}$-distill now underperforms $T$-scaling without further smoothing.

**Using an ensemble of $\mathcal{G}$-distilled networks**, at the sacrifice of test computational efficiency. Specifically, we train a bagging ensemble using standard training procedures, obtain ensemble soft labels $\widetilde{\mathbf{y}}_{\mathcal{F}}$ and $\widetilde{\mathbf{y}}_{\mathcal{G}}$ like before, but train 10 $\mathcal{G}$-distilled networks using the same set of soft labels (without resampling for convenience). At test time, the outputs from these networks are averaged to get the probability estimate. This scheme has the same drawback as the ensemble – reduced test efficiency. We name this scheme "$\mathcal{G}$-distill $\times 10$". For completeness, we also compare to an ensemble of the standard distillation, "distilling $\times 10$", which may already behave well on novel data due to model averaging.

As shown in Figure 8, we find that for $\mathcal{G}$-distill $\times 10$, the foreign NLL improves beyond the original ensemble, while the familiar NLL still falls slightly behind. Considering smoothing trade-off for all methods, $\mathcal{G}$-distill $\times 10$ still fall behind an ensemble. Our E99 confident error rates on both familiar and novel data are usually similar to or lower than the ensemble and distilling $\times 10$. Our accuracy or mAP is on the same level as the ensemble, except for Figure 8(c). These indicate that without $\tau$ smoothing, the $\mathcal{G}$-distill ensembles are better behaved on novel samples than the original ensembles, although the former tend to be less confident with familiar data.

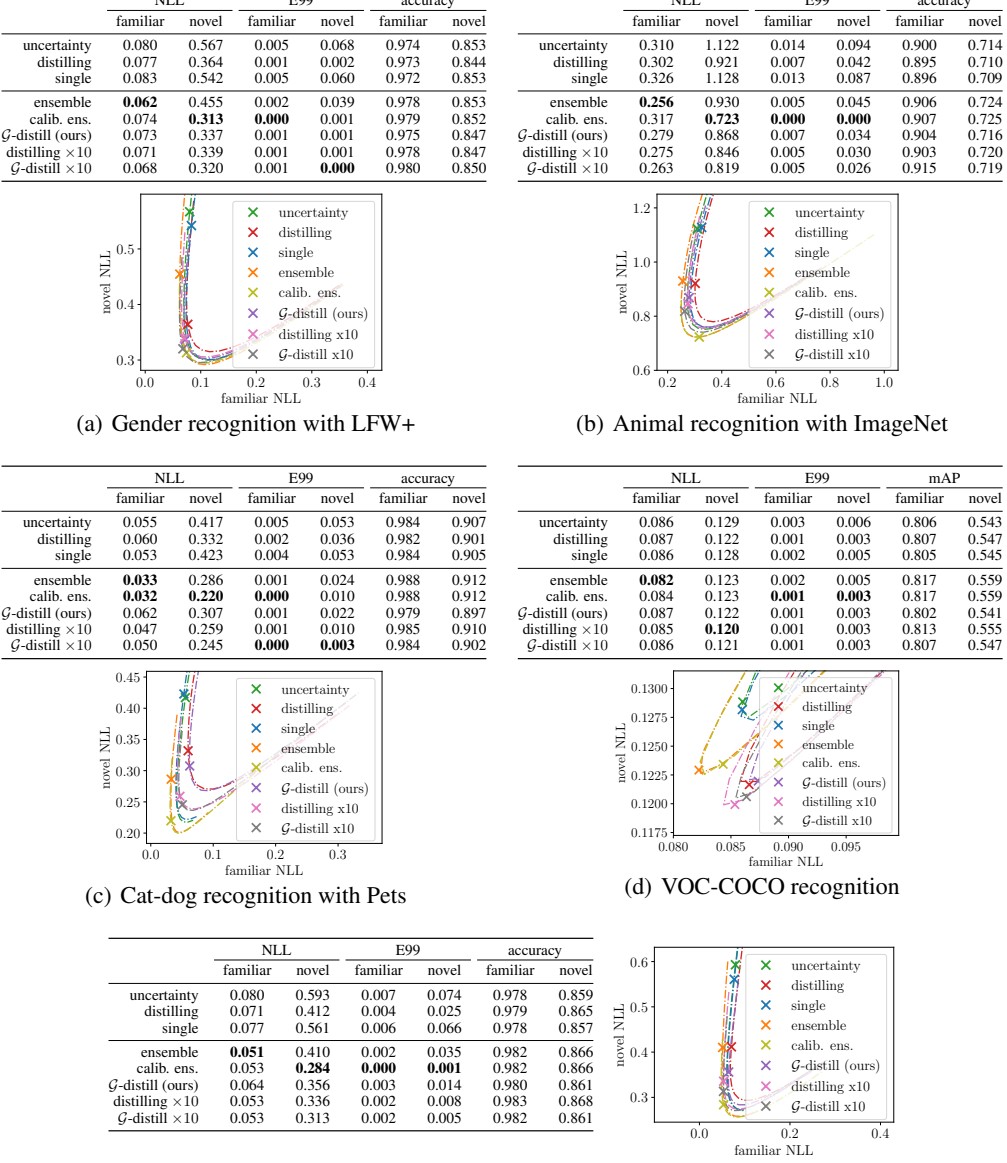

|  | NLL | | E99 | | accuracy | |
|---|---|---|---|---|---|---|
|  | familiar | novel | familiar | novel | familiar | novel |
| uncertainty | 0.080 | 0.567 | 0.005 | 0.068 | 0.974 | 0.853 |
| distilling | 0.077 | 0.364 | 0.001 | 0.002 | 0.973 | 0.844 |
| single | 0.083 | 0.542 | 0.005 | 0.060 | 0.972 | 0.853 |
| ensemble | **0.062** | 0.455 | 0.002 | 0.039 | 0.978 | 0.853 |
| calib. ens. | 0.074 | **0.313** | **0.000** | 0.001 | 0.979 | 0.852 |
| $\mathcal{G}$-distill (ours) | 0.073 | 0.337 | 0.001 | 0.001 | 0.975 | 0.847 |
| distilling ×10 | 0.071 | 0.339 | 0.001 | 0.001 | 0.978 | 0.847 |
| $\mathcal{G}$-distill ×10 | 0.068 | 0.320 | 0.001 | **0.000** | 0.980 | 0.850 |

(a) Gender recognition with LFW+

|  | NLL | | E99 | | accuracy | |
|---|---|---|---|---|---|---|
|  | familiar | novel | familiar | novel | familiar | novel |
| uncertainty | 0.310 | 1.122 | 0.014 | 0.094 | 0.900 | 0.714 |
| distilling | 0.302 | 0.921 | 0.007 | 0.042 | 0.895 | 0.710 |
| single | 0.326 | 1.128 | 0.013 | 0.087 | 0.896 | 0.709 |
| ensemble | **0.256** | 0.930 | 0.005 | 0.045 | 0.906 | 0.724 |
| calib. ens. | 0.317 | **0.723** | **0.000** | **0.000** | 0.907 | 0.725 |
| $\mathcal{G}$-distill (ours) | 0.279 | 0.868 | 0.007 | 0.034 | 0.904 | 0.716 |
| distilling ×10 | 0.275 | 0.846 | 0.005 | 0.030 | 0.903 | 0.720 |
| $\mathcal{G}$-distill ×10 | 0.263 | 0.819 | 0.005 | 0.026 | 0.915 | 0.719 |

(b) Animal recognition with ImageNet

|  | NLL | | E99 | | accuracy | |
|---|---|---|---|---|---|---|
|  | familiar | novel | familiar | novel | familiar | novel |
| uncertainty | 0.055 | 0.417 | 0.005 | 0.053 | 0.984 | 0.907 |
| distilling | 0.060 | 0.332 | 0.002 | 0.036 | 0.982 | 0.901 |
| single | 0.053 | 0.423 | 0.004 | 0.053 | 0.984 | 0.905 |
| ensemble | **0.033** | 0.286 | 0.001 | 0.024 | 0.988 | 0.912 |
| calib. ens. | **0.032** | **0.220** | **0.000** | 0.010 | 0.988 | 0.912 |
| $\mathcal{G}$-distill (ours) | 0.062 | 0.307 | 0.001 | 0.022 | 0.979 | 0.897 |
| distilling ×10 | 0.047 | 0.259 | 0.001 | 0.010 | 0.985 | 0.910 |
| $\mathcal{G}$-distill ×10 | 0.050 | 0.245 | **0.000** | 0.003 | 0.984 | 0.902 |

(c) Cat-dog recognition with Pets

|  | NLL | | E99 | | mAP | |
|---|---|---|---|---|---|---|
|  | familiar | novel | familiar | novel | familiar | novel |
| uncertainty | 0.086 | 0.129 | 0.003 | 0.006 | 0.806 | 0.543 |
| distilling | 0.087 | 0.122 | 0.001 | 0.003 | 0.807 | 0.547 |
| single | 0.086 | 0.128 | 0.002 | 0.005 | 0.805 | 0.545 |
| ensemble | **0.082** | 0.123 | 0.002 | 0.005 | 0.817 | 0.559 |
| calib. ens. | 0.084 | 0.123 | 0.001 | 0.003 | 0.817 | 0.559 |
| $\mathcal{G}$-distill (ours) | 0.087 | 0.122 | 0.001 | 0.003 | 0.802 | 0.541 |
| distilling ×10 | 0.085 | **0.120** | 0.001 | 0.003 | 0.813 | 0.555 |
| $\mathcal{G}$-distill ×10 | 0.086 | 0.121 | 0.001 | 0.003 | 0.807 | 0.547 |

(d) VOC-COCO recognition

|  | NLL | | E99 | | accuracy | |
|---|---|---|---|---|---|---|
|  | familiar | novel | familiar | novel | familiar | novel |
| uncertainty | 0.080 | 0.593 | 0.007 | 0.074 | 0.978 | 0.859 |
| distilling | 0.071 | 0.412 | 0.004 | 0.025 | 0.979 | 0.865 |
| single | 0.077 | 0.561 | 0.006 | 0.066 | 0.978 | 0.857 |
| ensemble | **0.051** | 0.410 | 0.002 | 0.035 | 0.982 | 0.866 |
| calib. ens. | 0.053 | **0.284** | **0.000** | 0.001 | 0.982 | 0.866 |
| $\mathcal{G}$-distill (ours) | 0.064 | 0.356 | 0.003 | 0.014 | 0.980 | 0.861 |
| distilling ×10 | 0.053 | 0.336 | 0.002 | 0.008 | 0.983 | 0.868 |
| $\mathcal{G}$-distill ×10 | 0.053 | 0.313 | 0.002 | 0.005 | 0.982 | 0.861 |

(e) Animal recognition with ImageNet, using DenseNet161

Figure 8: Using an ensemble of $\mathcal{G}$-distilled models to further boost the performance. Although we do obtain a better novel NLL compared to the ensemble, we usually lag behind the ensemble considering the smoothing trade-off.