# OpenReview forum: "Reducing Overconfident Errors outside the Known Distribution"
_ICLR.cc/2019/Conference_

### Official Review · AnonReviewer3 · 2018-10-27
**simple regularization reduces overconfidence**

**Rating:** 6
**Confidence:** 3

**Review:**

This paper introduces two methods of adjusting the overconfidence error for predictions on novel data. The ensemble distillation approach is to penalize the distillation loss on a potentially unlabeled general dataset.  The second approach (NCR) detects the novelty first and reweigh the prediction based on the familiarity to training data.

*stationarity*
From the statistical perspective, the overconfidence of extrapolation can kick in from two sources: a)the epistemic uncertainty.  The point estimation of softmax ignores the uncertainty of prediction at all.  A full Bayesian approach will remedy this though computationally impractical.  b) the generative distribution p(y|x) might not be identical on training data and test data.  To see the difference, if the training sample size goes to infinity, the uncertainty in a) will go to zero, but b) may still exist.  Section 3 assumes the invariant p(y|x) in novel data.  But theoretically, both methods do not require such invariance?

Slightly related here, there can be novel data for classification, and in principle, there can also be novel data for novelty detection?  That will make NCR fail.

*why the distillation helps uncertainty adjustment*
I am not convinced how the g-distillation works for this task.  In the extreme case if the ensemble model itself is totally wrong for novel data and the unlabeled general data used in training, how can I learn any extra uncertainty information from that noise? To be fair, when the temperature goes high enough, the ensemble will make uniform prediction and then the distillation loss is merely a loss function that enforces uniformity.  If I replace the ensemble softmax by a uniform prior for unlabeled general data, do I achieve the same effect?  That is essentially the same regularization as method 2, except g-distillation is on logit scale.

*robustness-accuracy tradeoff*
The experiments do not reveal too much robustness-efficiency conflict, as the new methods still perform good enough on familiar dataset. Indeed they can be even better than the baseline in E99 loss. Does it suggest the over-confidence is even a concern for familiar data/ iid data?

In general, the paper is well-written and well-motivated. It would be more interesting to make some theoretical explanation why/when this simple approach works.   I would recommend a weak accept at this point.

---

> ### Author Response · Authors · 2018-11-26
> **Rebuttal**
>
> Thank you for your feedback.
>
> - Important changes in rebuttal: please see our official comment.
>
> - Stationarity and what if ensemble fails: our method would not make sense if stationarity does not hold. In particular, our paper only helps reflecting epistemic uncertainty; we do not aim to address cases when the label generation process P(y|x) changes outside known distribution, because even ensemble methods would fail to provide a reasonable confidence on novel data.
>
> - Novel samples for novelty detection: Theoretically, there is no distinction between samples "novel for classifier" and "novel for novelty detector".
> Novelty detection methods like ODIN are usually designed to work on all possible novel samples, without any knowledge of the specific type of novel data they will be tested on, so they do not overfit to one type. Although due to e.g. hyperparameter selection, these methods may or may not generalize well, which may undermine our performance.
>
> - Why distillation helps:
> There seems to be a misunderstanding about enforcing uniformity -- we do not. As in Hinton et al., the high temperature is only used in training, not evaluation. For example, if the ensemble gives sample x a predicted distribution of p_Ens, and we use temperature T to make a smoother version p_EnsT, the optimization in training minimizes at the model outputting p_EnsT. But when evaluating, we use T=1 and the model would output p_Ens for that sample. Therefore we are only making the model output similar to the ensemble.
> One possible issue is if T is large, the model may get "lazy" and just outputs uniform for those samples and get stuck at a local minima, instead of learning to properly minimize the loss. This should not be an issue since our T=2 is quite small.
> Finally, NCR does have the effect that you mentioned, but it does not make novel samples' prediction completely uniform -- we use a lambda_s=0.15, according to a grid search on a validation split. Making it completely uniform (setting lambda_s=1) greatly undermines performance, so we think replacing softmax with uniform in G-distill would not work well either.
>
> - Robustness-accuracy tradeoff: the new methods sometimes suffer in the familiar distribution. We also observe that when we smooth the prediction linearly with the prior, we get better generalization but worse familiar performance (Figs. 3, 4).
> Over-confidence may become a concern if, for example, the training set is small and many samples from test data are in effect unseen during training. But we note that E99 is optimal anywhere in [0,0.01], since >99% confidence means there can be <1% predictions wrong. In familiar datasets, usually E99<0.01, with the worst case of 0.013 (Fig. 3b). So, E99 by itself does not indicate a over-confidence issue in familiar data. However, our further comparisons with a calibration method indicates some concern about over-confidence in familiar data.

---

> > ### Comment · AnonReviewer3 · 2018-12-01
> > **further comments**
> >
> > *misunderstanding about enforcing uniformity*  I was never saying your G-distillation enforces exact uniformity. I was saying the penalization term λ_dis L_dis can be thought as a prior that takes into account the novel unlabeled sample by the distillation, which by itself will prefer more uniform predictions compared with purely having the classification error (suppose hypothetically the label is known).   Of course, it will never be exact uniformity-- which is not desired at all. But it is more likely towards uniformity.
> >
> >
> > The new figures (3,4,6,7,8) are better illustrating robustness-accuracy tradeoff.

---

> > > ### Author Response · Authors · 2018-12-01
> > > **We agree with this comment**
> > >
> > > Regarding uniformity, that makes sense.  We may have misunderstood your original comment. The main idea behind using the unlabeled data for G-distillation is that it enables the distilled classifier to mimic the ensembles predictions in a larger domain than the training set, which may (as experiments support) improve predictions for novel samples.  Encouraging the distilled classifier to be overly uncertain for unlabeled samples may result in inappropriately low confidence estimates, especially if the unlabeled data contains samples within the classifier's domain (e.g. face examples for a gender classifier).

---

### Official Review · AnonReviewer2 · 2018-11-02
**A marginally novel method to estimate classification confidence on novel data distributions; Experimental results need to be more comprehensive and they are not conclusive enough.**

**Rating:** 4
**Confidence:** 4

**Review:**

The authors proposed two methods to deal with estimating classification confidence on novel unseen data distributions. The first idea is to use ensemble methods as the base approach that helps in identifying uncertain cases and then using distillation methods to reduce the ensemble into a single model mimicking behavior of the ensemble. They propose a generalization of this idea, that is to also perform distillation on a more generic unsupervised data distribution (than the supervised one that is used in training the ensemble). It is not clear whether this distribution should overlap with the novel distribution as a requirement or not. The second idea is to use a novelty detector classifier and weight the network output by the novelty score.
My major concern is that the comparison doesn't seem to be sufficiently comprehensive. The main method that is used to compare against is (Kendall & Gal, 2017), in which the main aim seems to be reducing uncertainty and improving generalization error under i.i.d. assumptions. This is different from the main focus of the paper, which is to better estimate classification confidence on novel data distributions. It seems that other approaches, such as "Calibration methods" (Guo et al. 2017) are better aligned with the focus of the paper, and should be considered instead.
My other concern is that the novelty seems to be marginal: either extending distillation methods in a very natural form, or weighting the network output using a novelty detector.

---

> ### Author Response · Authors · 2018-11-26
> **Rebuttal**
>
> We appreciate the crictical feedback by Reviewer 2.
>
> We agree that comparing to Guo et al. (2017) is an important experiment since we aim for a lower NLL. Thank you for pointing this out. We have tested the method and added the results to our paper. We note that calibration methods also operate on an i.i.d. assumption, without doing anything special on novel data.
>
> When adding the comparison to Guo et al. (2017), we found that this method has a similar, sometimes better, effect as our method. We have revised the experiment result section and related text in our paper to reflect this finding. We believe that even with this finding, our paper is still interesting and useful as a novel empirical study of a new but practical problem of generalizing confidence prediction into a unknown domain, and how methods from related fields work in this setting.
> Please also see our official comment on important changes to the paper.
>
> On the other hand, we argue that the current comparisons with epistemic uncertainty papers are still necessary and expected by readers. In prior work, people model epistemic uncertainty to produce properly confident prediction on the *entire* input space, not just i.i.d. data. Please see:
> (Section 5.1) Gal, Yarin, and Zoubin Ghahramani. "Dropout as a Bayesian approximation: Representing model uncertainty in deep learning." ICML 2016.
> (Section 5.2) Blundell, Charles, et al. "Weight uncertainty in neural networks." ICML (2015).
> ... where people model epistemic uncertainties to get better estimate of confidence outside the training region. Note the first paper is a prior version of Kendall & Gal (2017). Also, Kendall & Gal (2017) does evaluate their uncertainties on datasets unseen in training, although not using the criteria of NLL.
>
> Regarding general dataset overlapping with novel dataset: we argue that it cannot be a requirement -- we do not know what the novel dataset is beforehand, so overlapping should not be a condition for the general dataset. The idea is that the diversity and complexity of G will make the network generalize for any N. For example, experimentally, MSCOCO does not have many animal subclasses in ImageNet, but is able to improve animal superclass classification. Same for CelebA improving gender recognition result on 0-10 year olds. We have added this argument to the paper.

---

### Official Review · AnonReviewer1 · 2018-11-03

**Rating:** 6
**Confidence:** 4

**Review:**

The paper proposes two ideas for reducing overconfident wrong predictions:
- Method 1: “G-distillation” of an ensemble with extra unsupervised data
- Method 2: Novelty Confidence reduction (NCR) using novelty detector

The paper is well-written and was a pleasure to read. In particular, I really enjoyed reading the introduction and related work. My main concern is that some of the contributions claimed were already shown in previous work (see method 1 below for details), and the novelty feels a bit limited. That said, I like the simplicity of the method and think that the extensive experiments on a variety of datasets and architectures is useful to the community.

Method 1:
- The paper claims “Draw attention to a counter-intuitive yet important problem of highly confident wrong predictions when samples are drawn from a unknown distribution that is different than training” as one of the contributions. Note that previous work has already shown that single models are overconfident on unknown classes and ensembles are less overconfident, e.g. see Section 3.5 of the paper:
Simple and Scalable Predictive Uncertainty Estimation using Deep Ensembles
https://arxiv.org/pdf/1612.01474.pdf
- If I understand correctly, the key difference is that the proposed method 1 also uses ensemble prediction on unlabeled data for distillation, which could make the distilled model more robust. Ensembling on unlabeled data for robustness does seem novel to me, however, the text needs to be updated to clarify the novelty.


Method 2:
- By off-the-shelf, do you mean a pre-trained network released by ODIN? Or did you train ODIN-based novelty detector on your dataset?
- There was a recent paper that proposed to reduce confidence on novel inputs, which might be worth discussing:
Reliable Uncertainty Estimates in Deep Neural Networks using Noise Contrastive Priors
https://arxiv.org/pdf/1807.09289.pdf


Minor issues:
- Figures 3,4 are a bit small and hard to see

---

> ### Author Response · Authors · 2018-11-26
> **Rebuttal**
>
> Thank you for your feedback and related work proposal.
>
> - Important changes in rebuttal: please see our separate official comment.
>
> - Regarding similar findings in previous work:
> Although we cite the "Simple and Scalable ... Ensemble" paper in our work, we realize a lack of explicit comparison. Our findings are in a different setting from prior work. Prior work demonstrates overconfidence in samples that are *semantically different* (e.g. NotMNIST is alphabets, and CIFAR is objects, compared to MNIST digits in training) and users expect *uniform* prediction, which the network may fail to deliver. Our work further finds overconfidence on samples that are *semantically identical* but from an unseen *sub*category (e.g. male or female from different age groups) and users expect *sharp and correct* prediction, which the network may also fail to deliver.
> We argue that this distinction is not only far from trivial, but also points to a more practical but less studied scenario -- deep models meet unexpected data that users reasonably think they can process, but the models cannot do it properly, instead of data that nobody would reasonably expect the models to process. We will revise the paper to distinguish our findings from the related work.
>
> In terms of methodology, G-distill aims at *compressing* the ensemble into a single model, while preserving the behavior on out-of-distribution samples. This makes G-distill faster at test time, and potentially better regulated on novel. We argue this in the intro and related work, but we will make it clearer by mentioning the paper in the intro.
>
>
> - Regarding off-the-shelf rejector ODIN:
> Sorry for the confusion. ODIN is a model-free procedure(*) that does not involve any additional rejector network. We use ODIN with one of the recommended hyperparameters in our experiments. We will clarify this in Section 3.
>
> - Regarding the recent confidence reduction method:
> Thank you for the pointer. The paper is related to NCR, and has a similar out-of-distribution passive learning experiment at the end. We will discuss it in related work. We note that they perform regression on a low-dimensional feature space (and generalize to future data), which is a very different problem from our image classification (and generalize to unexpected samples). They use perturbed original dataset features and the prior as the labels for reducing confidence. This can be hard to generalize to an 224x224x3 image space.
>
> - Regarding Figure 3,4: we will increase the font size of the figures.
>
>
> (*) Roughly, ODIN (1) takes the classifier and a test input, (2) try to construct an adversarial example of the test input with one gradient step, (3) gauge how successful the adversarial example construction is, and (4) use this as the criteria to determine whether the test input is from a distribution seen in training. No additional network or training is involved.

---

### Public Comment · (anonymous) · 2018-11-10
**Questions about existing distillation method**

Hi, do you have any result comparing G-distillation with [1] and [2]? In [1], they also proposed using unlabeled data for training.

[1]  Bayesian Dark Knowledge (https://arxiv.org/abs/1506.04416)
[2] Distillation Dropout  (http://proceedings.mlr.press/v48/bulo16.pdf)

---

> ### Author Response · Authors · 2018-11-26
> **Thanks for pointing this out!**
>
> Thanks for pointing this out! We note that distillation dropout [2] is an approximation of MC-dropout (it does not outperform MC-dropout), and MC-dropout is a prior version of the uncertainty paper (Kendall & Gal, 2017) that we mainly compare to, so we should perform better than distillation dropout.
> For Bayesian Dark Knowledge, we apologize that we have not had the time to compare to this method, but it uses gaussian noise to generate out-of-sample data. This may work for low-dimensional data and MNIST, but we think this probably will underperform on a 224x224x3 image space.

---

### Author Response · Authors · 2018-11-26
**Important changes in the paper during rebuttal**

Reviewer 2 suggested comparison with Guo et al. (2017), a calibration method designed for deep networks for a better NLL in familiar test data. When performing the comparison, we found that this method has a similar, occasionally better, effect as our method on novel data as well. We have revised the experiment result section and related text in our paper to reflect this finding. We believe that even with this finding, our paper is still interesting and useful as a novel empirical study of a new but practical problem of generalizing confidence prediction into a unknown domain, and how methods from related fields work in this setting. We have modified the text to lay more emphasis on this aspect of our contribution.

In addition, during this investigation, we found that our way of smoothing prediction for the novel-familiar trade-off figure (taking weighted average between prediction and prior) is inferior to raising the temperature of the logits (similar to Guo et al.), so we changed the way we plotted the graphs for Figures 3,4,6,7,8.

---

### Meta-Review · Area_Chair1 · 2018-12-12
**Limited novelty**

**Confidence:** 4
**Recommendation:** Reject

**Metareview:**

The paper proposes methods to deal with estimating classification confidence on unseen data distributions.

The reviewers and AC note the following potential weaknesses: (1) limited novelty and (2) the authors' new comparison with Guo et al. (2017) asked by Reviewer 2 is not convincing enough.

AC thinks the proposed method has potential and is interesting, but decided that the authors need new ideas to meet the high standard of ICLR.